# Risk Factors for Surgical Site Infection after Soft-Tissue Sarcoma Resection, Including the Preoperative Geriatric Nutritional Risk Index

**DOI:** 10.3390/nu10121900

**Published:** 2018-12-03

**Authors:** Hiromi Sasaki, Satoshi Nagano, Noboru Taniguchi, Takao Setoguchi

**Affiliations:** 1Department of Orthopaedic Surgery, Graduate School of Medical and Dental Sciences, Kagoshima University, Kagoshima 890-8520, Japan; piro422@m2.kufm.kagoshima-u.ac.jp (H.S.); naga@m2.kufm.kagoshima-u.ac.jp (S.N.); tanigu@m2.kufm.kagoshima-u.ac.jp (N.T); 2Department of Medical Joint Materials, Graduate School of Medical and Dental Sciences, Kagoshima University, Kagoshima 890-8520, Japan

**Keywords:** soft-tissue sarcoma, surgical site infection (SSI), geriatric nutritional risk index (GNRI), malnutrition

## Abstract

Malignant soft-tissue sarcoma resection is associated with a relatively high incidence of surgical site infection (SSI). The known risk factors for SSI following soft-tissue sarcoma resection include tumor size and location, prolonged surgery, and massive blood loss. The geriatric nutritional risk index (GNRI) was used as a tool to help predict the occurrence of SSI after major surgery. We investigated the utility of the GNRI as a predictor of SSI following soft-tissue sarcoma resection. We retrospectively reviewed 152 patients who underwent surgical resection of soft-tissue sarcoma in our institute, and found that the incidence of SSI was 18.4% (28/152). The SSI and non-SSI groups significantly differed regarding surgical time, diameter of the skin incision, maximum tumor diameter, instrumentation, presence of an open wound, preoperative chemotherapy, preoperative C-reactive protein concentration, and GNRI. Binomial logistic regression analysis showed that the risk factors for SSI following soft-tissue sarcoma surgery were male sex, larger skin incision diameter, larger maximum tumor diameter, presence of an open wound, and lower GNRI. Our findings indicate that malnutrition is a risk factor for SSI after soft-tissue sarcoma resection, and suggest that appropriate assessment and intervention for malnutrition may reduce the incidence of SSI.

## 1. Introduction

Soft-tissue sarcomas are a rare and heterogeneous group of tumors that account for 1% of all adult malignancies, affect almost every site in the body, and retain the full range of malignant behavior [1]. The incidence of surgical site infection (SSI) following tumor resection is high, and is especially high in cases of high-grade sarcoma, such as undifferentiated pleomorphic sarcoma and liposarcoma [2]. Because of its rarity, the risk factors for SSI after soft-tissue sarcoma resection are yet to be well clarified.

Patients with cancer reportedly have high rates of malnutrition (40–80%) [3]. The strong correlation between cancer and inflammation is well known [4]. Systemic inflammation in patients with cancer causes an elevation in C-reactive protein (CRP), and decreases in serum albumin and total protein, which reflects malnutrition [5]. Thus, preoperative nutritional intervention is important for patients with cancer [6]. However, malnutrition is often unrecognized because of ineffective screening techniques [7]. Malnutrition in patients with cancer is related to poorer clinical outcome, poor quality of life, and poor prognosis [8]. Malnutrition is also associated with a higher incidence of SSI, higher incidence of morbidity, and longer duration of hospitalization following major surgery [9].

The geriatric nutritional risk index (GNRI) was generated to evaluate the risk of malnutrition-related complications in adult patients [10], and is a significant predictor of prognosis in many types of cancer [11,12,13,14,15,16,17,18]. The present study aimed to evaluate the potential risk factors for SSI, including preoperative GNRI, following soft-tissue sarcoma resection [19].

## 2. Materials and Methods

### 2.1. Patient Data

We retrospectively examined the records of 152 patients who were treated for soft-tissue sarcoma at the Department of Orthopedic Surgery, Kagoshima University from January 2007 to December 2016. Patients’ clinical characteristics were collected from the medical records, including sex, age, date of surgery, routine preoperative blood-test results, tumor size, tumor location, treatment, and comorbidities. An open wound was defined as a skin defect that remained after tumor resection and required secondary closure via skin grafting. Hypertension was defined in accordance with the World Health Organization/International Society of Hypertension guidelines as a blood pressure greater than 140/90 (grade 1) [20]. The results of preoperative blood tests including white-blood-cell count, hemoglobin concentration, CRP concentration, total protein, and total cholesterol were extracted to evaluate the preoperative nutritional status. The occurrence of SSI was assessed in accordance with the definition of the Centers for Disease Control and Prevention [21]. Patients for whom some of these data were missing were excluded from the study.

### 2.2. Geriatric Nutritional Risk Index

The GNRI was calculated from the serum albumin concentration and bodyweight using the following formula: GNRI = (1.489 × albumin (g/L)) + (41.7 × (bodyweight/ideal body weight)). The bodyweight/ideal bodyweight value was set to 1 when the patient’s bodyweight exceeded the ideal bodyweight [10]. The ideal bodyweight was defined as a body mass index (BMI) of 22 kg/m^2^ [12,22].

### 2.3. Statistical Analysis

Patients were divided into those who developed SSI (SSI group) and those who did not (non-SSI group). Differences in variables between the SSI and non-SSI groups were evaluated using the Student’s *t*-test, Mann–Whitney U test, and Fisher’s exact test. Correlation coefficients were analyzed via Spearman’s rank correlation coefficient. When the correlation coefficients between variables were >0.6, only one variable with the incidence of SSI, i.e., skin incision, was selected. Multivariable stepwise binomial logistic regression analysis was used to examine the relationships between the incidence of SSI and the assessed variables. Because of the small number of patients and the relatively large number of variables, we applied a stepwise selection method to identify significant variables, as previously described [23]. A *p*-value <0.05 was regarded as significant. Analysis was performed using the BellCurve for Excel add-in software (Social Survey Research Information Co., Ltd., Tokyo, Japan).

### 2.4. Ethics Approval and Consent to Participate

The present study protocol was approved by the institutional review board of Kagoshima University (approval number 180033), and was in accordance with the 1964 Helsinki declaration and its later amendments or comparable ethical standards. All patients provided written informed consent for the publication of their medical data.

## 3. Results

The clinical and demographic characteristics of the 152 patients who underwent surgical resection of soft-tissue sarcoma are shown in Table 1. The histological types of sarcoma are shown in Table 2. The sarcomas were histologically graded in accordance with the Fédération Nationale des Centers de Lutte Contre le Cancer (FNCLCC) grading system (Table 3) [24]. The locations of the sarcomas are shown in Table 4.

The incidence of SSI was 18.4% (28/152). The following variables significantly differed between the SSI and non-SSI groups: surgical time, diameter of the skin incision, maximum tumor diameter, instrumentation, presence of an open wound, preoperative chemotherapy, preoperative CRP concentration, and preoperative GNRI (Table 5).

Binomial logistic regression analysis showed that the risk factors for SSI following soft-tissue sarcoma resection were male sex, larger diameter of the skin incision, larger tumor diameter, presence of an open wound, and lower GNRI (Table 6).

## 4. Discussion

In the present study, the incidence of SSI after soft-tissue sarcoma resection was 18.4%, and the preoperative GNRI differed significantly between the SSI and non-SSI groups, as the SSI group had a low nutritional status compared with the non-SSI group. In addition, we found that the two groups significantly differed regarding surgical time, diameter of the skin incision, maximum tumor diameter, instrumentation, presence of an open wound, preoperative chemotherapy, and preoperative CRP concentration; these variables may be risk factors for SSI following soft-tissue tumor resection. Our logistic regression analysis showed that the combination of sex, diameter of the skin incision, maximum tumor diameter, presence of an open wound, white-blood-cell count, preoperative GNRI, and ischemic heart disease had the highest coefficient of determination. These findings indicate that using the preoperative GNRI in combination with other variables can improve the prediction of SSI following soft-tissue sarcoma surgery.

Of the variables identified in the present study as significant predictors of SSI after soft-tissue sarcoma resection, the presence of an open wound and the GNRI are modifiable. Thus, clinicians should consider closing the tumor resection wound and improving the perioperative nutritional status to prevent SSI. The tumor resection wound may be closed via reconstructive surgery involving plastic surgery techniques, while the nutritional status may be improved via nutritional intervention. The incidence of malnutrition is reportedly high in not only patients with cancer, but also in patients with sarcomas, and some nutritional scores reflect the prognosis of these patients [7,8,25,26]. Malnutrition in patients with cancer results from catabolic alterations including inadequate nutritional intake, muscle protein depletion, and systemic inflammation caused [6]. The European Society for Clinical Nutrition and Metabolism (ESPEN) guidelines for surgery in patients with cancer (2017) state that the success of the surgery depends on not only technique, but also on perioperative nutritional interventions [27]. In particular, the ESPEN guidelines emphasize the importance of perioperative nutritional interventions in preventing SSI in patients with cancer [27]. Nutritional therapeutic interventions for cancer-associated malnutrition include counseling, oral nutritional supplements, artificial nutrition, drug therapy, and physical therapy. All patients with cancer should undergo preoperative screening for malnutrition, and substrate and energy requirements should be met by stepwise nutritional interventions, from counseling to parenteral nutrition [6]. More than 70 nutritional assessment tools were reported in different populations [28]. Although nutritional screening is recommended, a fully sensitive and specific nutritional assessment tool is yet to be established [29]. As the high incidence of malnutrition in patients with soft-tissue sarcoma is correlated with poor prognosis [30], nutritional intervention may promote not only a reduction in the risk of SSI, but also an improvement in the prognosis.

## 5. Conclusions

Our findings suggest that the preoperative GNRI is a simple and useful tool for predicting the risk of SSI following soft-tissue sarcoma resection. The use of complementary nutritional therapies to improve the GNRI may reduce the incidence of SSI.

Our study has several limitations. Firstly, this was a single-center cohort study; thus, selection bias might have occurred. Our findings should be confirmed in a multicenter study. Secondly, we examined relatively few patients and variables; a bigger cohort is needed to precisely evaluate the risk factors for SSI after soft-tissue sarcoma resection.

## Figures and Tables

**Table 1 nutrients-10-01900-t001:** Demographic data of patients with soft tissue sarcoma.

Variables	
Females	76/152 (50%)
Age at surgery (years)	64 (51–73)
Surgical time (min)	213 (146–307)
Diameter of the skin incision (cm)	20 (15–30)
Maximum tumor diameter (cm)	70 (47–110)
Deep tumor location	98/152 (64.5%)
Instrumentation	12/152 (7.9%)
Presence of an open wound after tumor resection	42/152 (27.6%)
Preoperative chemotherapy	18/152 (11.8%)
White-blood-cell count (/μL)	5625 (4445–6638)
Hemoglobin concentration (g/dL)	13.3 (11.8–14.5)
C-reactive protein concentration (mg/dL)	0.14 (0.04–0.99)
Total protein (g/dL)	7.2 (6.9–7.5)
Total cholesterol (mg/dL)	195.2 ± 38.2
Geriatric nutritional risk index	104.2 (97.0–108.7)
Hypertension	50/152 (32.9%)
Ischemic heart disease	9/152 (5.9%)
Diabetes mellitus	19/152 (12.5%)

**Table 2 nutrients-10-01900-t002:** Types of sarcomas.

Pathological Type	Number
Undifferentiated pleomorphic sarcoma	50
Liposarcoma	33
Atypical lipomatous tumor	19
Synovial sarcoma	8
Myxofibrosarcoma	15
Leiomyosarcoma	6
Rhabdomyosarcoma	3
Others	18
Total	152

**Table 3 nutrients-10-01900-t003:** Fédération Nationale des Centers de Lutte Contre le Cancer grading of the sarcomas.

FNCLCC Grading	Number
Grade 1	19
Grade 2	38
Grade 3	81
Unknown grade	14
Total	152

**Table 4 nutrients-10-01900-t004:** Locations of the sarcomas.

Location	Number
Upper extremity	19
Trunk	33
Lower extremity (thigh)	100 (85)
Total	152

**Table 5 nutrients-10-01900-t005:** Comparison of patients with versus without surgical site infection (SSI) after resection.

Variables	SSI (+)	SSI (−)	*p*-Value
Number of patients	28	124	
Proportion of females	35.7% (10/28)	53.2% (66/124)	0.142
Age at surgery (years)	68 (56–74)	64 (51–73)	0.193
Surgical time (min)	284 (211–447)	198 (140–284)	0.001 *
Diameter of the skin incision (cm)	30 (20–40)	20 (15–30)	0.005 *
Maximum tumor diameter (cm)	98 (60–160)	69 (41–100)	0.010 *
Deep tumor location	57.1% (16/28)	63.5% (81/124)	0.514
Instrumentation	17.9% (5/28)	5.6% (7/124)	0.049 *
Proportion of patients with an open wound after tumor resection	46.4% (13/28)	22.7% (29/128)	0.019 *
Preoperative chemotherapy	25.0% (7/28)	8.9% (11/124)	0.026 *
White-blood-cell count (/μL)	5685 (4088–7460)	4458 (4458–6513)	0.994
Hemoglobin concentration (g/dL)	12.9 (11.9–14.5)	13.3 (11.8–14.6)	0.341
C-reactive protein concentration (mg/dL)	0.70 (0.10–2.99)	0.12 (0.04–0.73)	0.014 *
Total protein (g/dL)	7.0 (6.7–7.4)	7.3 (6.9–7.5)	0.092
Total cholesterol (mg/dL)	188.5 ± 31.2	196.7 ± 39.6	0.181
Geriatric nutritional risk index	99.8 (92.3–105.5)	104.3 (98.3–108.9)	0.026 *
Hypertension	39.3% (11/28)	31.5% (39/124)	0.505
Ischemic heart disease	14.3% (4/28)	4.0% (5/124)	0.060
Diabetes mellitus	14.3% (4/28)	12.1% (15/124)	0.754

* *p* < 0.05. The differences in variables between the SSI and non-SSI groups were evaluated using the Student’s *t*-test, Mann–Whitney U test, and Fisher’s exact test.

**Table 6 nutrients-10-01900-t006:** Binomial logistic regression analysis of the risk factors for surgical site infection (SSI) after resection.

Variables	Hazard Ratio (95% Confidence Intervals)	*p*-Value
Female sex	0.458 (0.171–1.225)	0.120
Diameter of the skin incision (cm)	1.045 (1.005–1.087)	0.028 *
Tumor diameter	1.007 (1.001–1.014)	0.029 *
Presence of an open wound after tumor resection	4.420 (1.547–12.627)	0.006 *
White-blood-cell count (/µL)	1.000 (1.000–1.000)	0.051
Geriatric nutritional risk index	0.951 (0.908–0.996)	0.034 *
Ischemic heart disease	3.933 (0.774–19.99)	0.099

Multivariable stepwise binomial logistic regression analysis was used. * *p* < 0.05.

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
