# Peer review of "Risk Factors for Surgical Site Infection after Soft-Tissue Sarcoma Resection, Including the Preoperative Geriatric Nutritional Risk Index"

_nutrients, 2018, doi:10.3390/nu10121900_

Reviewer 1 Report

The abstract seems to be interesting and adequately made. It explains the variables, objectives and the findings of the study. However, it is possible that it is not completely related to the title because is claimed GNRI is a unique predictor for SSI, which is clearly associated with another risk factors. Thus, I suggest review the title in order to include all the risk factors associated with SSI. A proposal would: "Risk factors and GNRI related to SSI in the soft tissue sarcomas resection.

Introduction

Line 32. The authors indicate different types of malignancies, so I suggest specifying the kind of sarcoma.

Please review the same paragraph and as possible avoid using the term soft tissue sarcomas too many times.

Material and methods

Please explain what you are referring to with the variables with the highest correlation.

Results

The authors should specify what they are referring to with the presence of open wound (before the surgery, after to the surgery or if it is caused by the tumor).

The authors generalized in the tables the high blood pressure of all the patients. Please establish adequately the stages of the high blood pressure that are related to SSI.

Please review the term "larger maximum tumor diameter".

The results of the tables offer interesting data. However, I consider that the authors should indicate at the figure legend what type of statistical tests were made regarding the significant results, because the correlation studies are not clear at all.

Despite of the fact that results are significant and interesting, I suggest to complement them with the next proposals:

1. Would it be possible to indicate the type of sarcomas that were evaluated?

2. Would it be possible to associate FNCLCC grading system of sarcomas and histological grade (G1, G2, G3 / G4) with the GNRI?

3. Why were not evaluated patients treated with QT, RT or RT / QT post-surgery and the GNRI? As well, would it be possible to explain why was only the QT variable taken in account before the surgery?

4. Would it be possible to make a survival analysis with Kaplan-Meier, in which could be considered GNRI in association with sarcoma and overall survival, disease-free survival and disease-specific survival?

5. Would it be possible to indicate at the figure legend the statistical tests that were used?

6. I consider important that, as possible, the authors make a table in which specify the characteristics of each evaluated patient.

Discussion.

In the first paragraph of this section (lines 121-125) the authors give an overall explanation of the surgical procedures and the SSI in sarcomas. First, I consider that it must be included an explanation associated to the results. Moreover, the word “malignant” has to be eliminated of “malignant soft tissue sarcoma”.

Were all the surgeries made of orthopaedical type? Why do the authors highlight this type of surgery more than others?

Please review lines 127-131 in order to establish the association of this paragraph to the obtained results in this study.

Please review lines 132-137 because is repetitive regarding to the introduction, which could be deleted. Moreover, this section must be more focused in the analysis of GNRI in sarcomas.

Lines 137-146: the authors make a correct inclusion of the results in this section, which could be improved establishing a comparison to other studies with similar assessed variables in order to present a stronger discussion.

Please review the term “systematic inflammation” and include its meaning.

Overall the discussion is interesting. However, the authors should focus it in the obtained results indeed. In other words, they have to compare the assessed variables with previous studies and establish proposals regarding to the results of their study.

Lines 168-171. As possible, include the limitations in the conclusions.

Author Response

The abstract seems to be interesting and adequately made. It explains the variables, objectives and the findings of the study. However, it is possible that it is not completely related to the title because is claimed GNRI is a unique predictor for SSI, which is clearly associated with another risk factors. Thus, I suggest review the title in order to include all the risk factors associated with SSI. A proposal would: "Risk factors and GNRI related to SSI in the soft tissue sarcomas resection.

As per your advice, we have revised the title to make it clear that GNRI is not the sole predictor of SSI after soft tissue sarcoma resection. The new title is “Risk factors for surgical site infection after soft tissue sarcoma resection, including the preoperative geriatric nutritional risk index”. Alternatively, we could change this to “Preoperative geriatric nutritional risk index as one of the risk factors for surgical site infection after soft tissue sarcoma resection”.

Introduction

Line 32. The authors indicate different types of malignancies, so I suggest specifying the kind of sarcoma.

We inserted a table in the Results section that lists the types of sarcoma.

Please review the same paragraph and as possible avoid using the term soft tissue sarcomas too many times.

We modified the paragraph to make it easier to read.

Material and methods

Please explain what you are referring to with the variables with the highest correlation.

Thank you for your comment. We added the following sentence in the Materials and methods section: The variable that had the highest correlation with the incidence of SSI was skin incision (correlation coefficient: 0.2288).

Results

The authors should specify what they are referring to with the presence of open wound (before the surgery, after to the surgery or if it is caused by the tumor).

We added an explanation of what was meant by ‘open wound’ as follows:An open wound was defined as a skin defect that remained after tumor resection and required secondary closure via skin grafting.”

The authors generalized in the tables the high blood pressure of all the patients. Please establish adequately the stages of the high blood pressure that are related to SSI.

We added the definition of high blood pressure as follows: “Hypertension was defined in accordance with the World Health Organization/International Society of Hypertension guidelines as a blood pressure greater than 140/90 (grade 1) [20].

Please review the term "larger maximum tumor diameter"

Thank you for your comment. We changed the phrase from ‘larger maximum tumor diameter’ to ‘larger tumor diameter’.

The results of the tables offer interesting data. However, I consider that the authors should indicate at the figure legend what type of statistical tests were made regarding the significant results, because the correlation studies are not clear at all.

We indicated which statistical tests were used in the table legend.

Despite of the fact that results are significant and interesting, I suggest to complement them with the next proposals:

1.     Would it be possible to indicate the type of sarcomas that were evaluated?

We inserted a table listing the types of sarcoma that were evaluated in the Results section.

2. Would it be possible to associate FNCLCC grading system of sarcomas and histological grade (G1, G2, G3 / G4) with the GNRI?

We inserted a table listing the histological grades of the sarcomas in the Results section.

3. Why were not evaluated patients treated with QT, RT or RT / QT post-surgery and the GNRI? As well, would it be possible to explain why was only the QT variable taken in account before the surgery?

We agree that postoperative chemotherapy and radiation may influence the development of SSI. However, the aim of the present study was to evaluate whether the preoperative nutritional status of patients with soft tissue sarcoma affected the incidence of SSI, and to evaluate the relationship between preoperative malnutrition and the occurrence of SSI. Thus, we only considered preoperative chemotherapy as a variable.

4. Would it be possible to make a survival analysis with Kaplan-Meier, in which could be considered GNRI in association with sarcoma and overall survival, disease-free survival and disease-specific survival?

We previously reported that malnutrition was associated with poor prognosis in patients with soft tissue sarcoma (reference #28). In that previous study, we evaluated nutritional scores such as the Glasgow prognostic score (GPS), GNRI, neutrophil-lymphocyte ratio, platelet-lymphocyte ratio, and controlling nutritional status; some scores significantly differed between patients who had a poor outcome versus those with a good outcome, and GPS (but not GNRI) was a prognostic factor for patients with soft tissue sarcoma (reference #28).

We discussed the relationship between malnutrition and prognosis in patients with soft tissue sarcoma in the Discussion section as shown below.

“As the high incidence of malnutrition in patients with soft tissue sarcoma is correlated with poor prognosis [28], nutritional intervention may promote not only a reduction in the risk of SSI, but also an improvement in the prognosis.”

5. Would it be possible to indicate at the figure legend the statistical tests that were used?

We stated which statistical tests were used in the table legend.

6. I consider important that, as possible, the authors make a table in which specify the characteristics of each evaluated patient.

We added three tables specifying the characteristics of each patient in the Results section.

Discussion.

In the first paragraph of this section (lines 121-125) the authors give an overall explanation of the surgical procedures and the SSI in sarcomas. First, I consider that it must be included an explanation associated to the results. Moreover, the word “malignant” has to be eliminated of “malignant soft tissue sarcoma”.

As per your advice, we deleted lines 121–125, as this information was not related to the present results.

Were all the surgeries made of orthopaedical type? Why do the authors highlight this type of surgery more than others?

We selected patients with soft tissue sarcoma, as the incidence of SSI after soft tissue sarcoma resection is higher than that after common orthopedic surgery procedures. As you mentioned, it would be interesting to investigate the relationship between the GNRI and SSI in other types of orthopedic surgery such as arthroplasty and spine surgery. The relationship between the GNRI and rheumatoid arthritis, and the relationship between the GNRI and hemodialysis have been reported previously (Nutrients. 2018 Feb 18;10(2), Asia Pac J Clin Nutr. 2018;27(5):1062-1066.).

Please review lines 127-131 in order to establish the association of this paragraph to the obtained results in this study.

As per your advice, we deleted lines 121–125, as this information was not related to the present results.

Please review lines 132-137 because is repetitive regarding to the introduction, which could be deleted. Moreover, this section must be more focused in the analysis of GNRI in sarcomas.

As per your advice, we deleted lines 132–137, as this information had already been stated in the Introduction section.

Lines 137-146: the authors make a correct inclusion of the results in this section, which could be improved establishing a comparison to other studies with similar assessed variables in order to present a stronger discussion.

As per your advice, we started the paragraph using these lines, as this information best reflects the present results.

Please review the term “systematic inflammation” and include its meaning.

Thank you for pointing out this mistake. We changed the word from ‘systematic’ to ‘systemic’.

Overall the discussion is interesting. However, the authors should focus it in the obtained results indeed. In other words, they have to compare the assessed variables with previous studies and establish proposals regarding to the results of their study.

As per your advice, we deleted the sentences that were not related to the present results, and revised the Discussion section.

Lines 168-171. As possible, include the limitations in the conclusions.

We included the limitations in the Conclusions section.

Reviewer 2 Report

In the manuscript by Sasaki et al., titled, "Geriatric nutritional risk index as a predictor of surgical site infection after soft tissue sarcoma resection," is a study that looks at a unique predictor in SSI of soft-tissue sarcoma. While others have looked at malnutrition in cancer recovery, it has not been studied in soft-tissue sarcomas and specifically within this patient population. 

The authors put forth compelling data as to why the geriatric nutritional risk index (GNRI) should be considered as a prognostic factor as to those patients who might develop SSI. This is especially relevant as this is a prognostic factor that can be changed pre- and post-operatively, as the authors stated in their discussion. 

In the discussed weaknesses, I would like to hear the authors arguments for and against using GNRI as the test for malnutrition. Are there other alternatives? And does it work in younger patient populations with soft-tissue sarcomas? 

Also in the description of the patient population, there are no details about the subtypes of sarcomas that were analyzed. Because of the heterogeneity of sarcomas, this information should be presented and tested to see if a specific subtype is overly represented in the patients with SSI. Along with subtypes, a general reference to anatomical location should be included (i.e. trunk vs. extremity) 

Stylistically, I would prefer to have tables with the first column left justified. Also in the paper there is no reference as to what the CRP stands for, so please reference this in parentheses the first time. Lastly, reference 22 is a duplicate of reference 19. Please remove one or the other. 

Author Response

In the manuscript by Sasaki et al., titled, "Geriatric nutritional risk index as a predictor of surgical site infection after soft tissue sarcoma resection," is a study that looks at a unique predictor in SSI of soft-tissue sarcoma. While others have looked at malnutrition in cancer recovery, it has not been studied in soft-tissue sarcomas and specifically within this patient population. 

The authors put forth compelling data as to why the geriatric nutritional risk index (GNRI) should be considered as a prognostic factor as to those patients who might develop SSI. This is especially relevant as this is a prognostic factor that can be changed pre- and post-operatively, as the authors stated in their discussion. 

We agree that the GNRI can change in accordance with the patient’s current status. As we wanted to evaluate the effect of preoperative nutritional status on the development of SSI, we changed the term from ‘GNRI’ to ‘preoperative GNRI’ throughout the present manuscript. However, we think it would be interesting to compare the effect of pre- and postoperative GNRI on the outcome of surgery in future.

In the discussed weaknesses, I would like to hear the authors arguments for and against using GNRI as the test for malnutrition. Are there other alternatives? And does it work in younger patient populations with soft-tissue sarcomas? 

Although many assessment tools have been used to evaluate nutritional status, no fully sensitive and specific nutritional assessment tool has yet been established.  We chose to use the GNRI to evaluate the presence of malnutrition in the present study. The GNRI was initially used to evaluate the nutritional status of older adult patients. However, GNRI is now used not only in older adult patients, but also in young patients to predict the outcome of many diseases such as cancer, hemodialysis, and coronary artery disease. No study has yet evaluated the relationship between preoperative GNRI and SSI in patients with soft tissue sarcoma.

Also in the description of the patient population, there are no details about the subtypes of sarcomas that were analyzed. Because of the heterogeneity of sarcomas, this information should be presented and tested to see if a specific subtype is overly represented in the patients with SSI. Along with subtypes, a general reference to anatomical location should be included (i.e. trunk vs. extremity) 

Thank you for your comments. As per your advice, we added three tables in the Results section that list the histological types, histological grades (FNCLCC), and locations of the sarcomas.

Stylistically, I would prefer to have tables with the first column left justified. Also in the paper there is no reference as to what the CRP stands for, so please reference this in parentheses the first time. Lastly, reference 22 is a duplicate of reference 19. Please remove one or the other. 

As per your suggestion, we changed the tables so that the first columns are left justified.

We also added the following information about the relationship between cancer and inflammation in the Introduction section: “The strong correlation between cancer and inflammation is well known [4]. Systemic inflammation in patients with cancer causes an elevation in C-reaction protein (CRP), and decreases in serum albumin and total protein, which reflects malnutrition [5].”

Thank you for pointing out our mistake in the reference list. We deleted reference #22.

Round  2

Reviewer 1 Report

Lines 73-75: Please review this section because the authors include a correlation coefficient with the number 0.2288, and according to the article made by Mukaka, M.M et al., Malawi Medical Journal;24(3):69-71,2012, the correlation coefficients << span="">.30 have a weak association. Thus, I would suggest to modify this text and delete the term “highest correlation” and the number 0.2288 in order to use only the term “correlation”.

Author Response

As per your advice, we modified this text as follows,

When the correlation coefficients between variables were > 0.6, only one variable with the incidence of SSI, skin incision, was selected.